# Posture Monitoring and Correction Exercises for Workers in Hostile Environments Utilizing Non-Invasive Sensors: Algorithm Development and Validation

**DOI:** 10.3390/s22249618

**Published:** 2022-12-08

**Authors:** Siavash Khaksar, Stefanie Pieters, Bita Borazjani, Joshua Hyde, Harrison Booker, Adil Khokhar, Iain Murray, Amity Campbell

**Affiliations:** 1School of Electrical Engineering, Computing, and Mathematical Sciences, Curtin University, Bentley, WA 6102, Australia; 2School of Allied Health, Curtin University, Bentley, WA 6102, Australia

**Keywords:** personal protective equipment (PPE), hostile environments, posture monitoring, musculoskeletal disorder, optical motion capture, fiber-optic sensing, e-textile sensors, inertial measurement unit (IMU), Kalman filter, Euler angles, quaternion angles, Unity, human joint measurement

## Abstract

Personal protective equipment (PPE) is an essential key factor in standardizing safety within the workplace. Harsh working environments with long working hours can cause stress on the human body that may lead to musculoskeletal disorder (MSD). MSD refers to injuries that impact the muscles, nerves, joints, and many other human body areas. Most work-related MSD results from hazardous manual tasks involving repetitive, sustained force, or repetitive movements in awkward postures. This paper presents collaborative research from the School of Electrical Engineering and School of Allied Health at Curtin University. The main objective was to develop a framework for posture correction exercises for workers in hostile environments, utilizing inertial measurement units (IMU). The developed system uses IMUs to record the head, back, and pelvis movements of a healthy participant without MSD and determine the range of motion of each joint. A simulation was developed to analyze the participant’s posture to determine whether the posture present would pose an increased risk of MSD with limits to a range of movement set based on the literature. When compared to measurements made by a goniometer, the body movement recorded 94% accuracy and the wrist movement recorded 96% accuracy.

## 1. Introduction

### 1.1. Background

PPE is an essential key factor in standardizing safety within the workplace. Harsh working environments with long working hours can cause stress on the human body may result in musculoskeletal disorder (MSD). MSD refers to injuries that impact the muscles, nerves, joints, and many other human body areas [1]. Most work-related MSD results from hazardous manual tasks involving repetitive, sustained force, or repetitive movements in awkward postures [1]. 

MSD impacts the workers and the employer in the form of economic loss due to absenteeism, lost productivity, increased health care, disability, and worker’s compensation claims [1]. Based on the Australian Workers’ Compensation Statistics from 2018 to 2019, 36% of compensation claims were due to body stress, resulting in a median of 6.2 weeks lost per severe claim [2]. The percentage rate of severe claims due to MSD between male and female workers is 87%, with laborers being the highest compared to several other working groups [2]. 

The age group most impacted by this issue are between 45 and 49 years of age. However, even the youngest workers under 20 years old have 3650 claims of injury and MSD [2]. These statistics show that this is, in fact, a severe issue that needs to be dealt with and will be beneficial for all working-age groups.

### 1.2. Existing Methods

A standard device currently used to measure joint angles is known as a goniometer. A specific type of goniometer is used to measure motion in the spine and is known as a gravity-dependent goniometer or inclinometer [3]. This method requires precision for an accurate reading that is only obtained through practice and skillful observation [3]. The slightest misplacement can lead to an inaccurate reading and usage would not be suitable in the proposed application area and will not offer continued monitoring of the active range of movement.

Safe Work Australia’s Hazardous manual task Code of Practice states that a movement that is repeated or sustained for long period that ranges 20° out of the human posture’s natural state can pose a significant risk of MSD [4]. An angle of 30° for spinal range is used to make the range less conservative. In addition to this, a goniometer is used to verify the obtained data.

Optical passive motion capture technologies use retro-reflective markers attached to the body parts of the individual that reflects light onto a nearby camera lens. From this reflection, the position of the marker is calculated within three-dimensional space and recorded [5]. This approach is also known as motion capture or mo-cap which is the process of digitally recording the movement of people [6]. This approach is used in sports, entertainment, ergonomics, medical applications, and robotics and is also known as performance capture when looking at the full body, face, and fingers. Optical active motion capture uses the same technique, but rather than reflecting light, the light is emitted [5]. Optical motion capture technology provided the most accurate results based on research [5] and is well equipped for use in a laboratory environment. This method is considered as the gold standard for capturing human movement; however, due to its considerable expense, with a simple Vicon system [7] costing around $250,000 Australian dollars in 2011 [8], its impracticality for small harsh environments, and its inherent complexity [9], optical motion sensing is impractical for most field-based settings. 

Fiber-optic sensors are another example of potential field use and rely on the measurement of light traveling through an optical fiber system. This measurement can be in terms of light intensity, phase, or polarization [10]. Fiber-optic sensing provided a robust design that could withstand harsh environments by tolerating high temperatures, offered a wide dynamic range and large bandwidth, and was not susceptible to electromagnetic interference, radio frequency, or corrosive environments [11]. Even though this is a new method recently developed for posture monitoring, it has shown that it is a solid competitor compared to optical motion capture technology producing similar results [12]. However, due to its considerable expense and inherent complexity, fiber-optic sensing was not chosen.

Another potential approach could be the use of e-textile sensors, which is a common phrase referring to electronic textiles. Electronic textiles are fabrics that incorporate electronics and interconnections woven within them [13]. E-textile sensors provided a less visible and invasive design. This method provided reliable results when compared to optical motion capture technology [14]. This procedure required minimal complexity to implement. Due to this method’s lack of durability in harsh environments (susceptible to interference with parasitic capacitance due to heavy sweating and relaxation of the tight stretchable fabric due to continuous use and washing) which can result in unreliable data, e-textile sensors were not chosen [14].

Inertial measurement units (IMU) are one of the popular field-based methods for tracking the movement and positioning of an object. IMU’s consist of an accelerometer to measure force and acceleration, a gyroscope to measure the rate of change in angles, and lastly a magnetometer that utilizes the earth’s magnetic field as a fixed reference for the current estimation of the IMU orientation to prevent drift [15]. 

The inertial measurement unit (IMU) provides a well-developed, non-invasive, affordable design with long battery life [16]. Less advanced theory is required to implement this method and has proven to be a reliable form of posture monitoring with several cases to refer to [17]. There is an option of customizing the IMU or choosing a pre- calibrated and developed system. Due to these advantages, IMUs were chosen as the desired method. There are several data-driven methods for using IMU data in conjunction with neural networks to classify human movement. For example, IMUs have previously been used for medical purposes such as capturing foot drop in [18] and the hand movement of children with cerebral palsy in [19]. Ref. [15] shows a system for using multiple IMUs connected to the legs of patients with foot drop issues and uses machine learning to classify the severity and need for surgery compared to healthy participants. Ref. [16] uses IMUs to capture the hand movement of children with cerebral palsy, as well as typically developing children, and uses machine learning to classify the movement associated with cerebral palsy from the IMU data. In another example, [20] provides a method for using image processing, neural networks, and public databases for capturing human movement. They implement this using 15 sensors. Even though the results look very promising, the large number of sensors and the processing power required to analyze the data are unsuitable for hostile environments. To overcome this issue, rather than relying on machine learning, the proposed system focuses on real-time quaternion data and a range of joint angle movements to monitor the user movement, as well as provide feedback to them for potential use in posture correction exercises. The detail of this implementation is explained in the methods section of the paper [20]. 

### 1.3. Contribution of the Paper

The main contribution of this paper is providing a digital, low-cost system and framework for human posture monitoring and exercises for workers in a hostile environment. The sensor set up provides a clinically accurate representation of wrist, elbow, and knee joint movement which has been validated in a Vicon motion analysis system and goniometers. This system and framework provide the ability to adjust the range of movement for different body parts and the length of time spent at each range. This framework can also be utilized as a digital rehabilitation tool where rehab exercises related to the wrist, elbow, and knee can be captured in real time and provide the user with feedback on how accurately the exercise is being completed. For this paper, the focus has been on providing this feedback for workers in a hostile environment. 

## 2. Methods

### 2.1. System Requirements

The aim of this paper is to provide a system that will enable workers to have their posture monitored whist doing certain activities and provide posture correction exercises, with feedback to the user so they can see if they are doing the exercise correctly. The target user will be any worker or individual that require having their posture monitored whilst doing activities. This system will be used to provide posture monitoring with a visual aspect for now.

Requirements that were identified as essential for the success and effectiveness of the project were that the system must be able to record live stream data, show a visual representation of movement, detect harmful and non-harmful angles which need to be defined for each user, and display a message for the user regarding the harmful position. Additional requirements that will be examined are the measurement of wrist, knee, and elbow joint angles.

A process diagram for the proposed system can be seen in Figure 1 and shows the process that was followed when implementing the system.

### 2.2. Selecting a Suitable Sensor

One of the first steps for this project was to select suitable IMU sensors that are small, lightweight, have a long battery life, utilize BLE 5.0 (Bluetooth Low Energy), have a high sampling rate, provide continuous measurements throughout the working shift, and are economically priced. BLE 5.0 connectivity is needed as it is more robust and can transmit 8 times more data at twice the speed compared to BLE 4.3 or BLE 2.1 [21]. 

There are a wide variety of off-the-shelf IMU sensors on the market; however, only a select few IMU sensors seen in Table 1 were used for comparison. With the aforementioned factors considered, Xsens Dot was chosen as the IMU to use in this project as it can be seen in Figure 2 that the Xsens Dot was the most well-rounded choice, although the Vicon blue trident was a close contender. However, due to the noticeable price difference Xsens Dot was chosen. Figure 3 shows a photo of the Xsens Dot sensors used in this paper. 

Orientation and free acceleration are obtained from the Xsens dot by means of an in-built fusion algorithm and a Kalman filter. This fusion algorithm is referred to as the XKFCore of the Xsens Dot IMU [22].

The Xsens Dot is sized at L:36.3 × W:30.4 × H:10.8 mm with a weight of 11.2 g. This provides a small and lightweight device that does not hinder the user’s movement. The internal storage is 64 MB with a sampling rate of 800 Hz. This provides enough storage for storing the captured data when necessary and a sampling rate capable of capturing fast movement. The output rate ranges from 1 Hz to 60 Hz with 120 Hz available only for recording. Communication is conducted through Bluetooth [22].

The Xsens Dot provides 9 h battery life which means the sensors can provide continuous motoring of the posture without the need to change the battery. The electrical current consumption of one Xsens Dot is 68 mA [22]. The battery within the Xsens dot is an LIR2032H rechargeable coin battery. Battery specifications include a nominal capacity of 70 mAh, a nominal Voltage of 3.7 V, and a working temperature of −20~+60° [22].

The Xsens Dot can operate in temperatures ranging from 0 to 50° Celsius which is within the required standards for underground environments. The IP Rating is IP68 which indicates that the Xsens Dot can withstand damage caused by dust or water (can be submerged up to 1.5 m deep) [26]. 

### 2.3. Filtering and Sensor Fusion

It is necessary to use fusion algorithms to filter out the external noise and integrate all the sensor data. There are several different methods used for filtering, namely, for example, Kalman filtering, complementary filtering, and particle filtering. The Xsens Dot uses an in-built fusion algorithm for capturing real-time orientation and a filtering method such as a Kalman filter.

Kalman filtering is one of the most common estimation algorithms and plays an essential role in the IMU fusion algorithm [27]. Developed in 1960, the Kalman filter is used today for navigation systems and control systems [28]. The objective of the Kalman filter is to minimize the mean squared error of the measured data compared to the estimated results [29]. This is completed by using two basic steps: prediction and correction. The prediction step uses the control commands given to predict where the dynamic system will be at the next point in time. The correction step uses the data obtained by the IMU sensor to correct any potential mistakes that have been made and determines a prediction error to use when the following prediction is made [27]. This prediction and correction step cycles continuously to provide accurate results and is known as recursive estimation. 

One limitation that the Kalman filter possesses is that it is not well suited for working with nonlinearities due to the assumptions made to develop a Kalman filter; however, since human movement is linear, it is not a major issue. These assumptions being that the filter will only work with Gaussian distribution, and all models are linear [30]. An alternative Kalman filter was created, known as the extended Kalman filter, that deals with non-linearities by performing local linearization with the Taylor approximation of the non-linear model to work around this problem. This method is used to turn it into a linear model based on linearization points that need to be updated for each prediction of the recursive estimation [30].

Euler angles describe the rotation and orientation of a body in three-dimensional space from an initial frame to a final frame [31]. The angles used are commonly known as yaw, pitch, and roll. Euler angles describe the orientation between two 3D coordinate systems. This orientation can be represented in a 3 × 3 coordinate system parameterized by Euler angles. 

Advantages to using Euler angles are that it is easier to visualize and can describe rotation and orientation in a precise manner [32]. Euler angles do have a disadvantage which is that this technique is susceptible to gimbal lock, which is the phenomenon where one degree of freedom is lost due to two axes aligning. For example, when the pitch approaches 90°, the roll and yaw is locked, thus making them indistinguishable [33]. Without an external reference, it is impossible to re-orientate the axis once gimbal lock occurs [33].

A method of working around gimbal lock is to use quaternion angles instead. Quaternion angles consist of 4 components: a real component and three imaginary components. Quaternion angles describe three-dimensional rotations and orientation with a generalization of complex numbers [32]. These angles are then later converted to a regular rational matrix, instead of a rotation matrix as seen with Euler angles. Quaternion angles simplify the equation by using a quaternion notation to represent a rotation of θ degrees about an axis defined by the vector $\hat{u}=\left(u_{x},u_{y},u_{z}\right)$, as seen in Equation (1).
(1)$$\hat{q}=\cos\left(\frac{\theta }{2}\right)+\left(u_{x}i+u_{y}j+u_{z}k\right)\sin\left(\frac{\theta }{2}\right)\;$$

### 2.4. Software Specification

The Unity game engine [34] and the Xsens Dot application were the tools utilized for implementing the software component of the posture monitoring framework.

Unity is a powerful system used for designing games and application scenes in 2D or 3D. With correct use of programming, Unity can be utilized to capture motion data for analysis. Programming language such as C# is used to develop scripts within the model. Unity was the main platform for developing the model as it has extensive reference and scripting documentation that can be used to start obtaining motion capture data as quickly as possible. 

The hardware Xsens Dot IMU provides an application called “Xsens dot” that is used to obtain motion capture data directly for the IMU via Bluetooth. This application has the capability to record real-time streaming of the IMU and log the data into a csv file that can be exported onto a computer for analysis. From the Xsens Dot application provided, the IMUs were connected to the application through Bluetooth connectivity. The advanced application gives the users the option to measure the quaternion, Euler, free acceleration, acceleration, magnetic field, and angular velocity.

### 2.5. Prototype Design

A prototype has been set up with a standard PPE helmet that has an inner frame for fitting adjustments (Figure 4). It was decided that the Xsens Dot sensor will be placed on the top of the head as this placement will provide the most accurate results. It was decided that a harness similar to the harness seen in [17] shall be created to determine the correct placement of the IMU on the user’s back for the best results. It was determined that placing the IMU sensors on the back of the chest and the hips is the best placement to receive reliable results. The harness prototype can be seen in Figure 5.

Harnesses similar to this are being used in some mines around the world. This harness is designed to carry additional load that would normally be placed around the belt/pants. This design prevents any possible injury or discomfort.

The Xsens Dot IMUs are placed within a plastic zip-lock bag and positioned on the harness with Velcro. This is not a permanent solution. However, it does provide a temporary solution to evaluate posture monitoring.

Utilizing the Xsens Dot application discussed, the head movement was monitored (validation of this IMU has already been completed and can be viewed in Validation). Figure 6 shows a user pivoting (bending) the head left and right to a 30° angle. Thirty degrees has been chosen as discussed in the existing Methods section. When the head pivots to the right, the Euler angle in the *X*-axis produced a positive 30° angle. When the head pivots to the left, the Euler angle in the *X*-axis produced a negative 30° angle. In the natural state the angle is nearly zero. The flexion and rotation of the head was also monitored and produced the same pattern of results with the *Y*-axis and *Z*-axis, respectively.

Figure 7 shows the participant pivoting (bending) the chest left and right to a 30° angle. When the chest pivots to the right, the Euler angle in the *X*-axis produced a positive 30° angle. When the chest pivots to the left, the Euler angle in the *X*-axis produced a negative 30° angle. In the natural state, the angle is nearly zero. The flexion and rotation of the chest was also monitored and produced the same pattern of results with the *Y*-axis and *Z*-axis, respectively.

Figure 8 shows the participant pivoting (bending) the hips left and right to a 30° angle. When the hips pivot to the right, the Euler angle in the *X*-axis produced a positive 30° angle. When the hips pivot to the left, the Euler angle in the *X*-axis produced a negative 30° angle. In the natural state, the angle is nearly zero. The flexion and rotation of the hips were also monitored and produced the same pattern of results with the *Z*-axis and *Y*-axis, respectively.

Standard Velcro straps were used to monitor any arm or leg movement. The orientation measurement of the arms and legs are not required as only the head, chest, and pelvis are the main body parts that will be monitored to determine poor posture. The Xsens Dot sensors will be positioned on the Velcro straps similarly by placing the Xsens Dot sensor into a plastic zip-lock bag and attaching the sensor to the arm or leg with Velcro.

### 2.6. Connecting to Sensors

The sensors needed to be connected to a computer via Bluetooth to transfer their data and pass them to Unity for visualization. To achieve this, a graphical user interface has been developed in Python that uses Bluetooth to scan for available IMUs, synchronizes them, and passes the information directly to Unity. The application has been developed by implementing a reliable Transmission Control Protocol (TCP) client (the Python Bluetooth module) and TCP receiver (Unity script asset called the ServerReceiver).

TCP (Transmission Control Protocol) is a transport layer protocol that is used in conjunction with IP to ensure the reliable transmission of packets. TCP is more reliable as it requires a handshake to start the session. Handshake refers to a connection establishment protocol where a connection request (CR) is first sent to the receiver from the sender and then waits for a ‘connection accepted’ sent to the sender from the receiver.

A GUI has been developed in Python where the users can scan the sensors, as seen in Figure 9. This interface streams the IMU orientation data to Unity where the data stream can be seen in Figure 10.

Once connection from all sensors have been established, selecting the run button will start streaming the quaternion angles from each sensor to the receiver. The TCP packets in JSON format are shown in Figure 10 where each line represents a new packet. With quaternion, ‘wq’ represents the real component, and the remainder represents the imaginary components.

### 2.7. Joint Angle Measurements Methods

Three different methods were examined to measure joint angle movement of the wrist, knee, and elbow. The method chosen needed to be able to replicate measurements that are made by a goniometer (a medical device used to measure joint angle during movement). The chosen method was incorporated within the simulation for users to access, if desired. The measurements made will also be accessible through a csv file within Unity. 

The first method involved obtaining two quaternion angles from the child game object and the object that it is immediately attached to (the parent). The quaternion angles obtained from the parent and child are used to create a rotate vector that is in reference to a unit vector on an axis (obtained by script quaternions.cs). A dot product between the two rotate vectors is obtained and arc cosine is used to produce an angle which is later converted to degrees. A high-level flow chart of the process (with wrist rotation used as an example) can be seen in Figure 11.

The second involved obtaining two quaternion angles from the child game object and the object that it is immediately attached to (the parent) and determining the difference in angle between the child body part and the parent body part. This difference is calculated by multiplying the inverse of the quaternion from the parent to the quaternion from the child. This new quaternion is converted to Euler and displayed to the user. A high-level flow chart of the process (with wrist rotation used as an example) can be seen in Figure 12.

The third method involved using a function provided by Unity, namely “gameobject.localRotation.eulerAngles”. This function provides an angle of the child game object in reference to the object that it is immediately attached to (the parent). For this example, this function will provide an angle, which can be represented as a joint angle, between the hand and the lower arm. A high-level flow chart of the process (with wrist rotation used as an example) can be seen in Figure 13. The three methods have been simultaneously compared to measurements made by a goniometer. 

### 2.8. Visulizing the Data in Unity 

The avatar used to mimic the participant’s movements was imported from the Unity asset store as it uses the ragdoll feature. This enables the user to develop a humanoid avatar with objects placed within their respective position based on the body mapping. With this helpful feature, it enables the user to have control of several joints of the humanoid with significant detail, as can be seen in Figure 14.

Further development of the scene was made, as seen in the result section to ensure users can see the date, time, movement angles of the main focused body parts, a message board when objects are out of bound, and a dropdown selection to change monitoring scenes. Further developments were made where joint angles can be monitored, thus a toggle selection for this method was incorporated as well. 

### 2.9. Sensor Evaluation 

The accuracy of the Xsens Dot IMU sensors were validated with Curtin University’s Vicon motion analysis lab. A simple wrist flexion and extension exercise was completed while the arm was resting on a table and reflectors were placed on the sensors.The data provided by the Vicon set up was then compared against the orientation data provided by the IMU. An example of the Vicon data vs. Xsens Dot data has been provided in Figure 15. Please note that the signals have been time shifted so the data can be compared more easily. The location of reflectors for the Vicon system can been in Figure 16. 

After the accuracy of the IMUs were validated, a goniometer was then used to compare the reported results from the sensors to the readings of the goniometer. In this validation, full body rotation and joint angle validation was compared to a goniometer. The results of this validation have been discussed in the preliminary findings section of this paper.

## 3. Results

Full body movement was made possible by developing an array of structs with 13 allocations. A struct is a collation of variables that can be different types in programming. Each allocation represents an essential body part used for posture monitoring. Each struct consists of a quaternion, three Euler angles for rotation, correction, and positional control, the Xsens Dot sensor number controlling their respected body object, and lastly the name of the object that is being controlled. Adil’s method was used to obtain the quaternion data of the Xsens Dot sensors. The quaternion angles of each sensor used are converted to Euler via an in-built Unity function called quaternion.eulerAngles [35]. Euler was chosen to be the displayed angle to the user as it is the easiest and most common angle used to understand rotation. OnAnimatorIK function [36] was used to provide the Euler angles to the avatar in order to create movement. This function that is provided by Unity gives the user the ability to access any object that is part of the avatar’s anatomy (seen in Figure 14). The final version of the environment can be seen in Figure 17 and the movement can be seen in Figure 18. Appendix A contain several video demonstrations for the framework. 

One problem that arose when implementing full body movement was that a calibration was needed to ensure that the avatar can return to in its natural state when the sensors set the avatar in an unnatural position. This calibration needed to be created in a way that still ensured that the angles provided were still deemed accurate and only needed to be completed the first time sensors were attached. This was achieved by creating a trigger token, the C button. When the user pressed the C button, the system will took a new reading of the Xsens Dot sensors and stored it in a temporary rotation float. From this, the original rotation is subtracted from the temporary rotation, providing a difference that establishes a new rotation between −180 and 180°. This method can be seen executed in Figure 19.

Once full body movement was attained, it was necessary to monitor different movement scenarios. It was determined that standing, sitting, lifting, and joint movement would be monitored in this model. A user interface dropdown selection was made that showcased the different scenarios available. This dropdown was connected to the setPosition.csv script. An integer variable between 0 and 4 was given from the dropdown list to the Pos function within setPosition.csv script representing scene None to Lifting, respectively. This function provided a Euler angle variable noted as “pos” to the Rotations.csv script. Through all 4 scenes, the head, chest, and hips are being monitored with the head, chest, hips, upper arms, and upper legs receiving rotation. Figure 20 showcases some natural movement that can be accomplished from selecting the sitting or lifting scene, respectively.

## 4. Discussion

### 4.1. Principal Findings 

As previously mentioned, three joint measurements methods were investigated. Method 1 and Method 2 stayed consistent, providing results similar to the measurements made by the goniometer, while Method 3 started producing promising results regarding the knee and elbow. Method 3 produced good results at a later state since the parent and child both started with a Euler angle of (0°, 0°, 0°). When the wrist joints were measured, the upper arm had a starting Euler angle of (−80°, 0°, 0°) and the lower arm had a starting angle of (0°, 0°, 90°). It was decided that Method 2 would be the method of choice as the results gave clear positive and negative values based on the choice of direction when completing the required motions. Method 2 also had clear singular changing X, Y and Z angles when completing each activity, which will be more favorable to the user.

The sensors were validated against a goniometer by taking angle measurements at 0, 10, 20, 30, 40, and 50° angles using the goniometers and comparing the readings with the IMU-based measurements. Table 2 illustrates the validation results of the head, chest, and hips when compared to measurements made by a goniometer. It is seen that the results can be deemed as reliable. Table 3 and Table 4 illustrate the validation results of a user’s wrist when the hand is in an ulnar and radial deviation, respectively. Table 5 and Table 6 illustrate the validation results of a user’s wrist when the hand is in flexion and extension, respectively. Table 7 and Table 8 illustrate the validation results of a user’s wrist when the hand is in pronation and supination, respectively. Table 9 showcases the validation results of a user’s knee when the hand is in flexion and extension. Table 10 showcases the validation results of a participant’s elbow when the hand is in flexion and extension. The results of the tests prove the reliability of Method 2 as the main source of joint measurement for this application. 

### 4.2. Conclusions

Upon completion of the project, an IMU-based human movement monitoring framework has been provided that can be expanded to various possibilities beyond posture monitoring. As described in the paper, this monitoring system relies on real-time quaternion data streamed via IMUs to Unity. Once the accuracy of the IMUs was validated against Vicon motion analysis set up at Curtin University, three separate joint angle measurements were implemented and validated against the goniometer. The goniometer comparison demonstrated Method 2 as being the most accurate for the application area. The most accurate method was achieved by obtaining two quaternion angles from the child game object and the object that it is immediately attached to (the parent), and determining the difference in angle between the child body part and the parent body part. This difference is calculated by multiplying the inverse of the quaternion from the parent to the quaternion from the child, converted to Euler and displayed to the user. A calibration functionality was also implemented since IMUs will demonstrate inherent drift overtime. 

There is scope for future work in the simultaneous joining of joint angle movement with full body posture monitoring such that rehabilitation exercises can be explored. Smoothness of motion can also be explored and included for further development in the accuracy of results that would be deemed useful in the medical field. One of the main advantages of the proposed system is that it does not rely on a specific type of IMU. As long as quaternion data can be read from the IMU, it can map to this framework. This has been made possible by moving all the calibration and joint angle measurement to the software. Additionally, since the software was developed in Unity, it can be easily ported to mobile platforms such as Android and Apple’s IOS and open the possibility of remote training where clinical staff can provide remote guidance and advice while the sensors are worn by the workers on site.

## Figures and Tables

**Figure 1 sensors-22-09618-f001:**
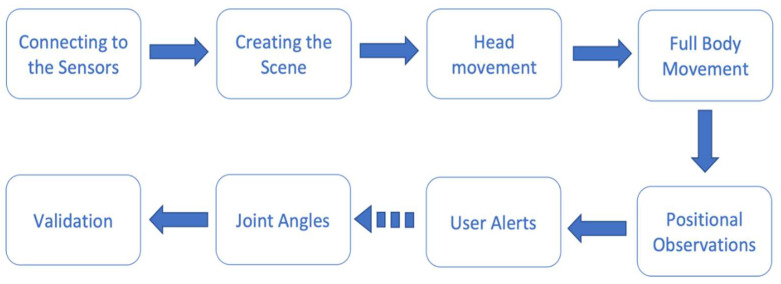
Process diagram.

**Figure 2 sensors-22-09618-f002:**
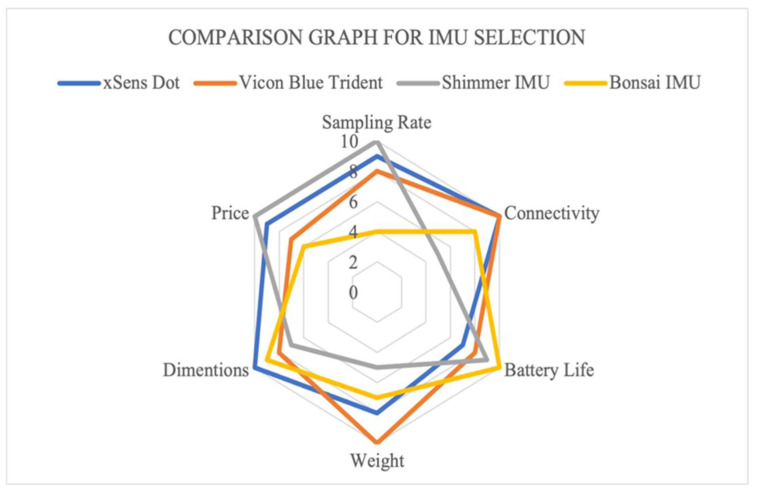
Comparison graph for IMU selection based on Table 1.

**Figure 3 sensors-22-09618-f003:**
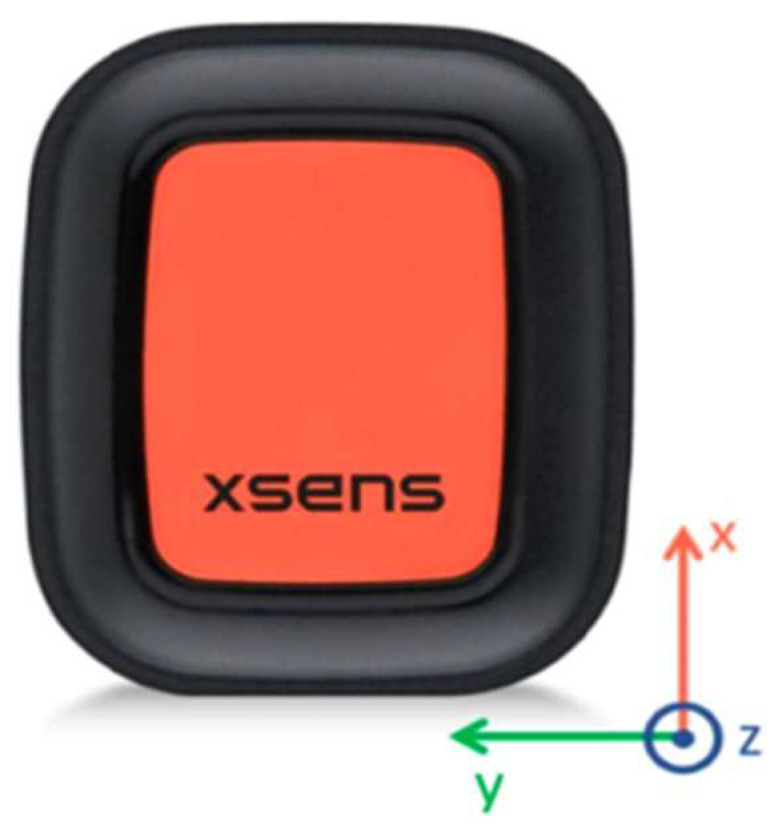
The Xsens Dot IMU [18].

**Figure 4 sensors-22-09618-f004:**
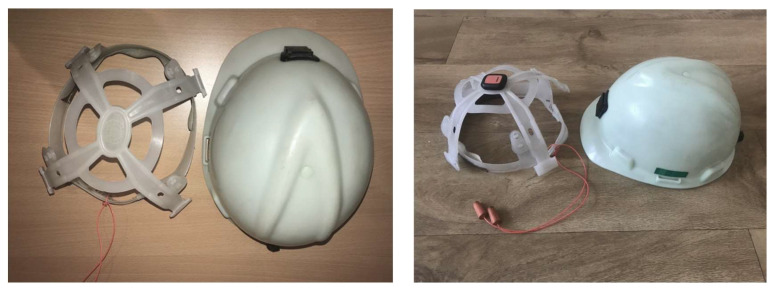
Helmet prototype setup.

**Figure 5 sensors-22-09618-f005:**
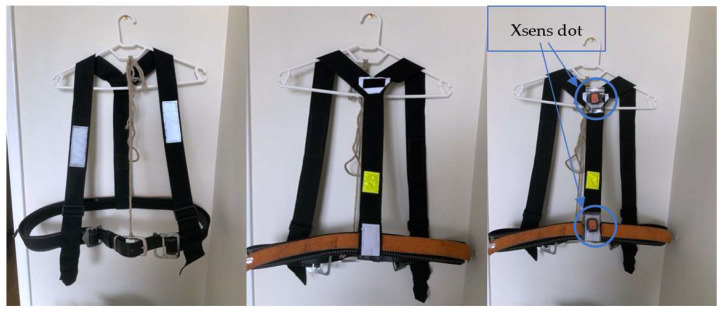
Harness prototype.

**Figure 6 sensors-22-09618-f006:**
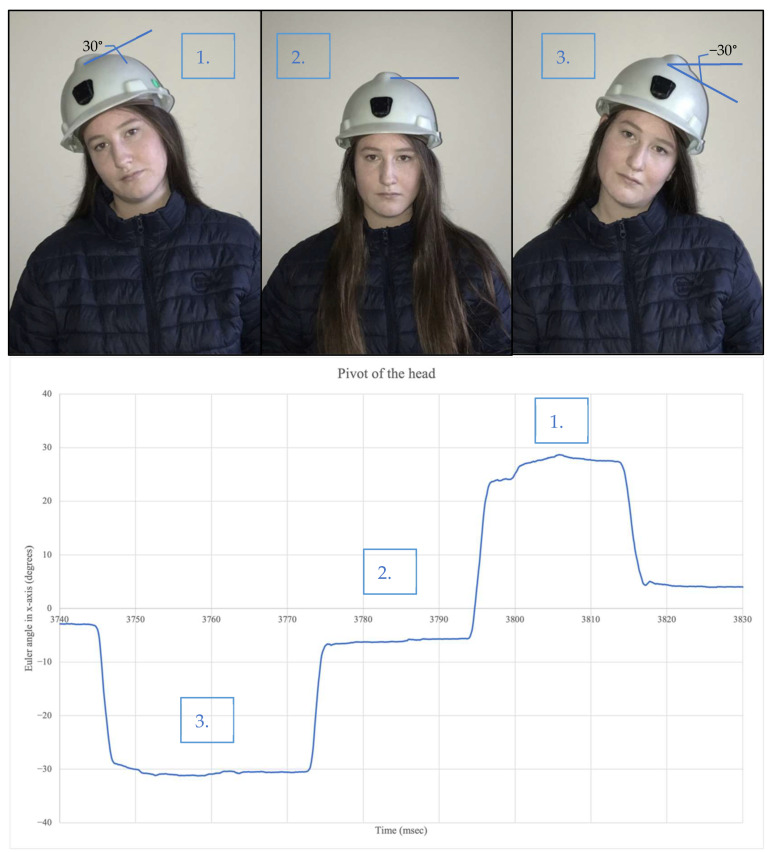
Orientation measurement with head pivoting.

**Figure 7 sensors-22-09618-f007:**
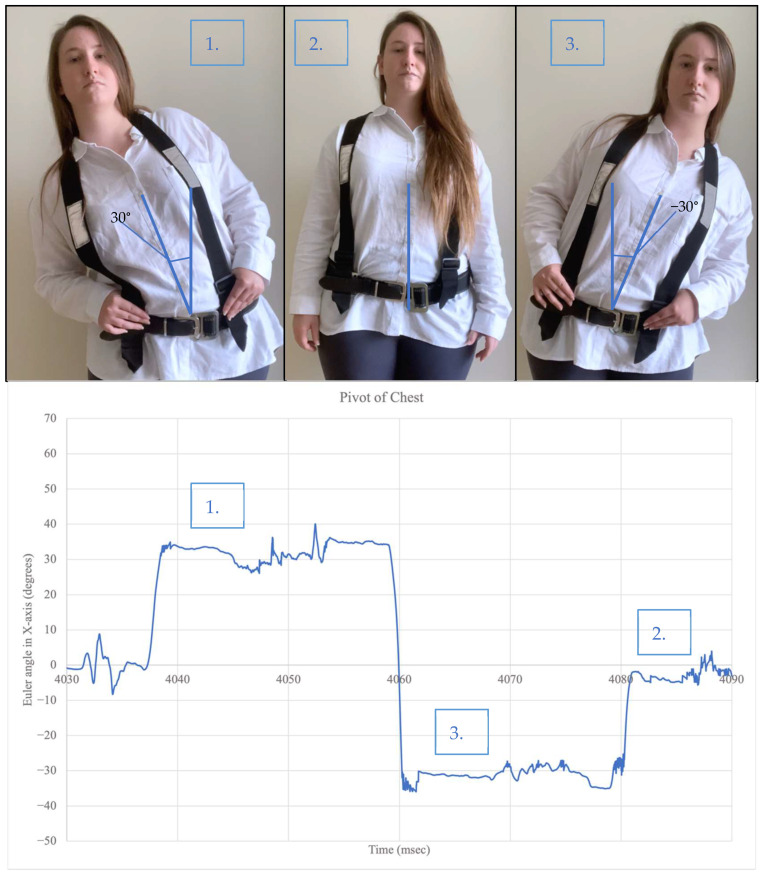
Orientation measurement with chest pivoting.

**Figure 8 sensors-22-09618-f008:**
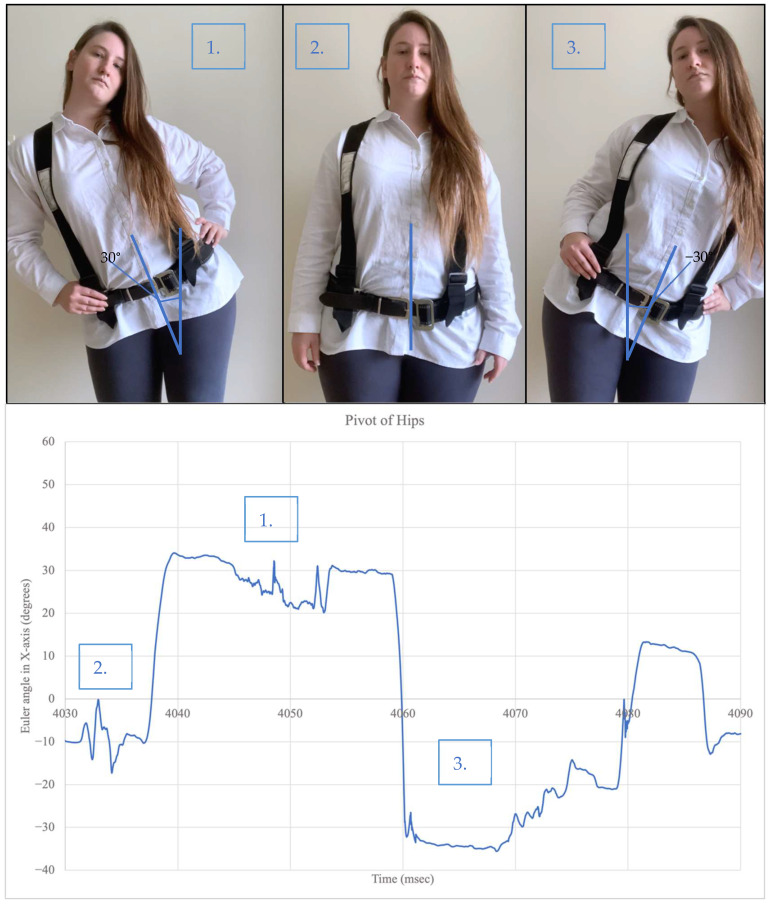
Orientation measurement with hips pivoting.

**Figure 9 sensors-22-09618-f009:**
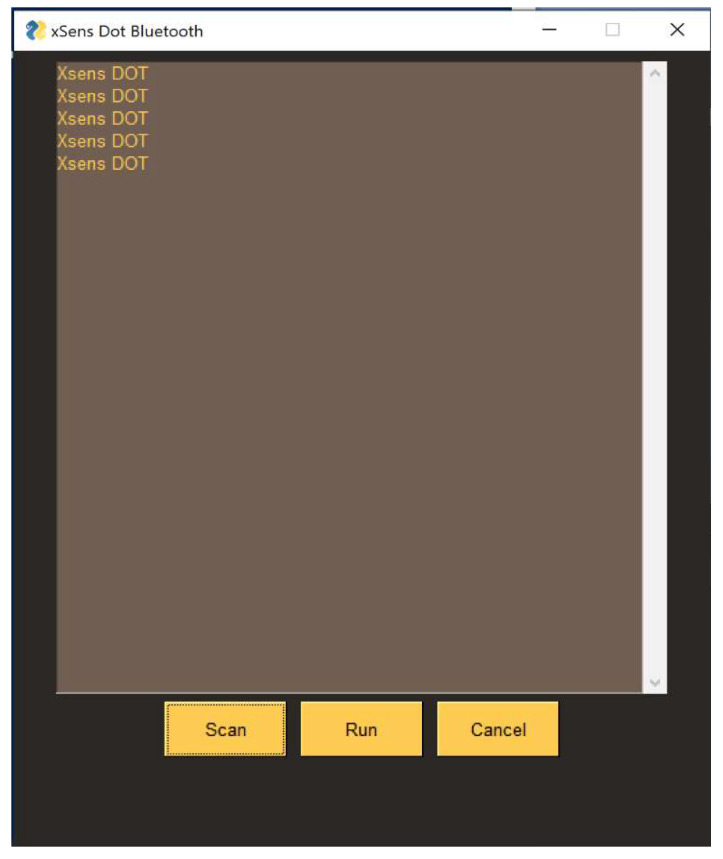
Bluetooth module and GUI.

**Figure 10 sensors-22-09618-f010:**
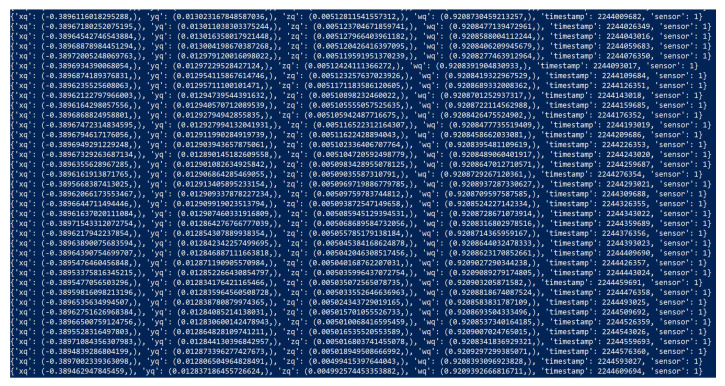
TCP Packets in JSON format.

**Figure 11 sensors-22-09618-f011:**
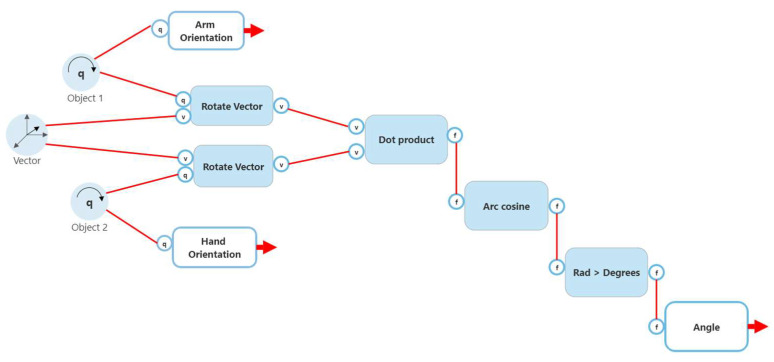
Method 1 joint calculation.

**Figure 12 sensors-22-09618-f012:**
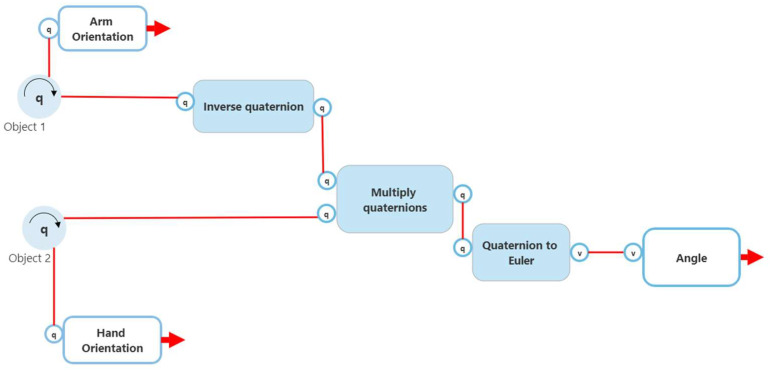
Method 2 joint calculation.

**Figure 13 sensors-22-09618-f013:**
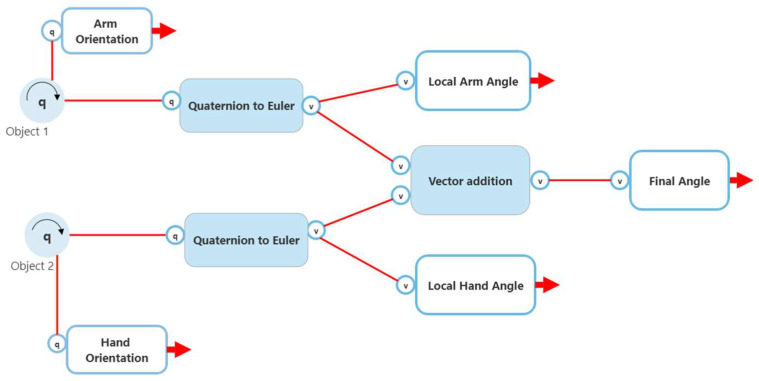
Method 3 joint calculation.

**Figure 14 sensors-22-09618-f014:**
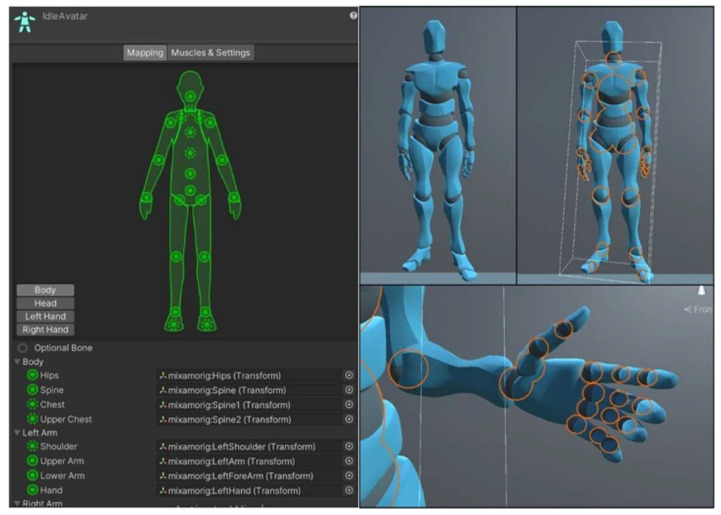
Body mapping of Unity avatar.

**Figure 15 sensors-22-09618-f015:**
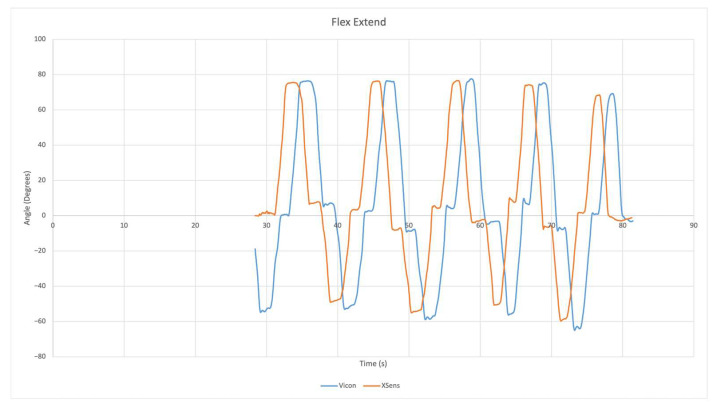
Validation of Xsens Dot IMU with Vicon.

**Figure 16 sensors-22-09618-f016:**
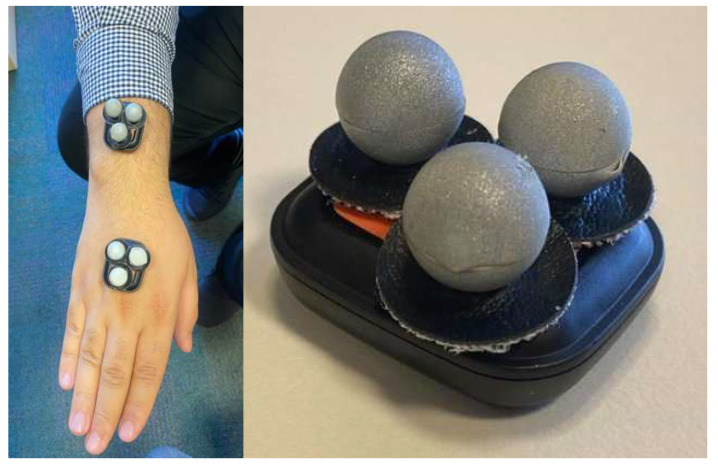
Placement of reflectors for the Vicon validation.

**Figure 17 sensors-22-09618-f017:**
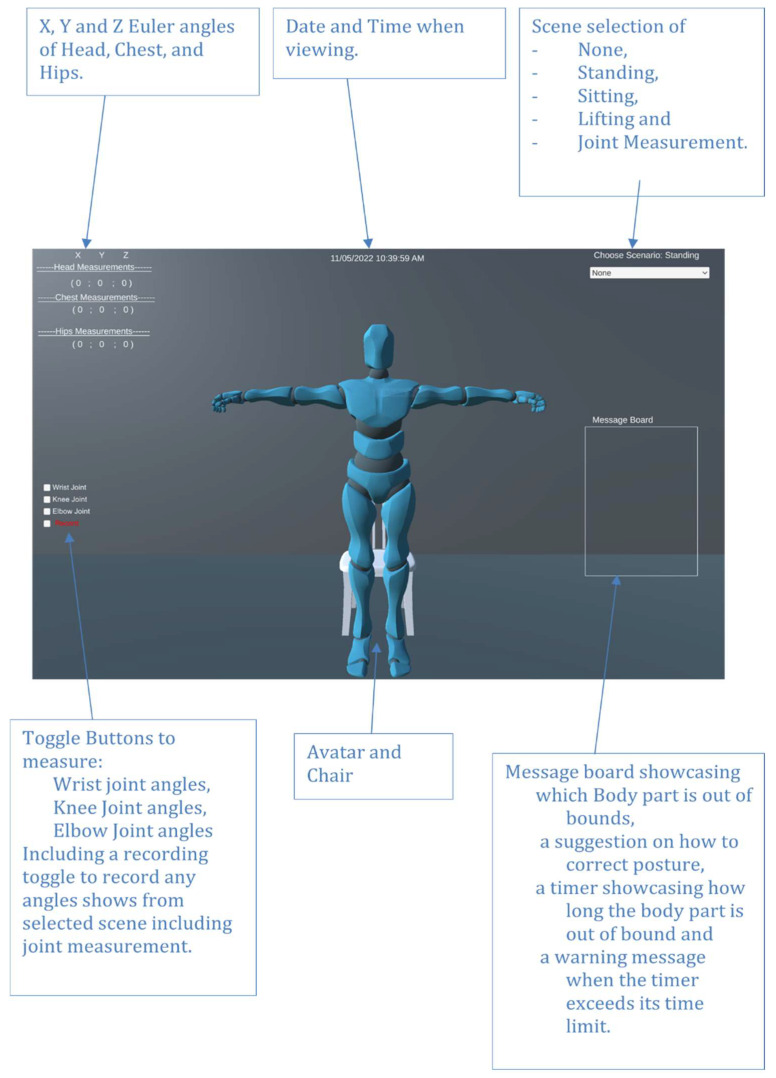
Scene prototype.

**Figure 18 sensors-22-09618-f018:**
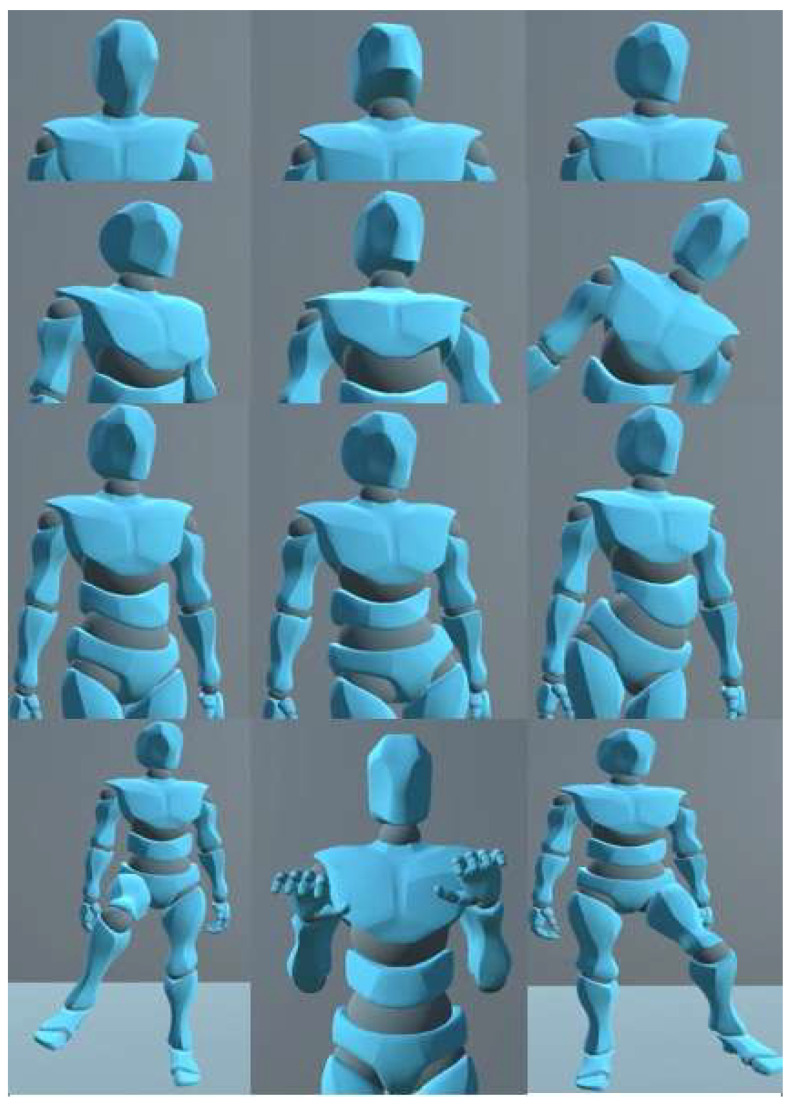
Full body movement of avatar.

**Figure 19 sensors-22-09618-f019:**
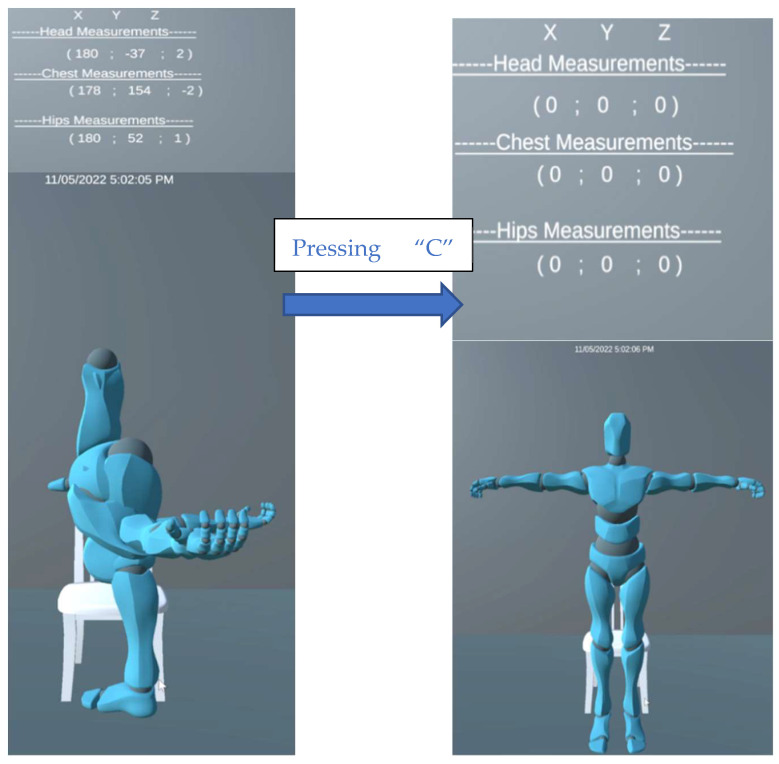
Implementing Calibration.

**Figure 20 sensors-22-09618-f020:**
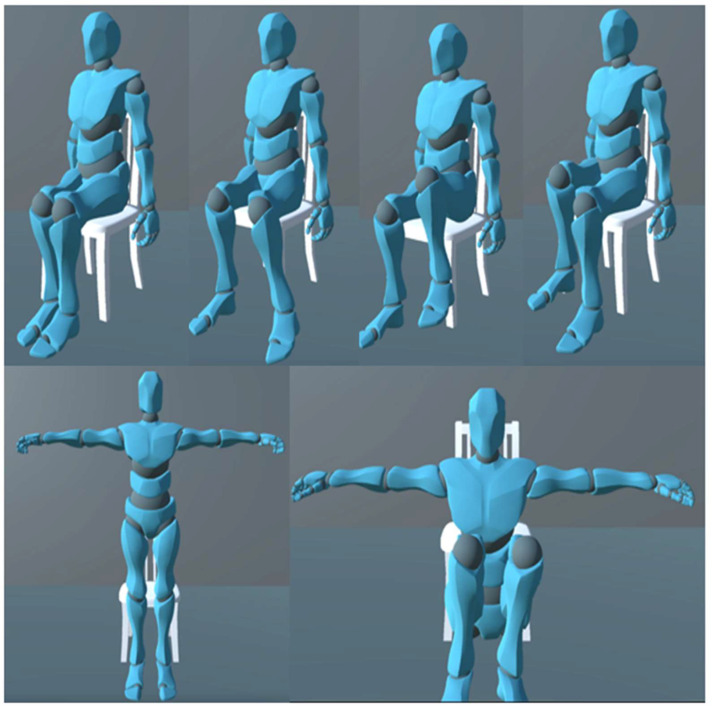
Natural sitting motion with sitting scene (**above**) and natural crouching motion with lifting scene (**below**).

**Table 1 sensors-22-09618-t001:** Different IMU choices for posture monitoring.

IMU	Sampling Rate	Connectivity	Battery Life	Weight	Size	Price
**Xsens Dot** [22]	120 Hz	BLE 5.0	9 h	11.2 g	36.3 × 30 × 10.8 mm	€495.00 (~$798.05 AUD) for 5 pack
**Vicon Blue Trident** [23]	100 Hz	BLE 5.0	12 h	9.5 g	42 × 27 × 11 mm	$1600.00 USD (~$2184.36 AUD) each
**Shimmer IMU** [24]	128 Hz	BLE 2.1	14 h	23.6 g	51 × 34 × 11 mm	€359.00 (~$578.79 AUD) each
**Bonsai IMU** [25]	50 Hz	BLE 4.3	16 h	15 g	36.5 × 32 × 13.5 mm	€2490.00 (~$4014.44 AUD) for 15 pack

**Table 2 sensors-22-09618-t002:** Measurements of full body movement (All angles are in degrees).

Goniometer	Head			Chest			Hips		
	Up and Down Motion	Left and Right Motion	Pivoting Left and Right Motion	Up and Down Motion	Left and Right Motion	Pivoting Left and Right Motion	Up and Down Motion	Left and Right Motion	Pivoting Left and Right Motion
Angle	X Angle	Y Angle	Z Angle	X Angle	Y Angle	Z Angle	X Angle	Y Angle	Z Angle
0	0	0	0	0	0	0	0	0	0
10	10	10	8	11	10	9	10	10	11
20	20	20	19	20	20	20	20	20	18
30	29	30	30	29	30	30	30	30	30
40	40	38	40	40	40	39	40	40	N/A
50	49	50	50	49	50	47	50	49	N/A

**Table 3 sensors-22-09618-t003:** Measurements of ulnar deviation of wrist (All angles are in degrees).

Goniometer	Method 1			Method 2			Method 3		
Joint Angle	X	Y	Z	X	Y	Z	X	Y	Z
0	0	0	1	−1	0	0	−7	−83	−35
10	11	NAN	11	−1	12	0	−1	−74	−36
20	19	NAN	19	−1	19	0	4	−67	−35
30	30	NAN	30	−1	31	0	11	−58	−34
40	38	NAN	38	−1	39	0	16	−49	−32
50	50	NAN	50	−1	52	0	21	−40	−29

**Table 4 sensors-22-09618-t004:** Measurements of radial deviation of wrist (All angles are in degrees).

Goniometer	Method 1			Method 2			Method 3		
Joint Angle	X	Y	Z	X	Y	Z	X	Y	Z
0	1	NAN	1	−1	1	0	−6	−82	−36
10	11	3	11	−1	−9	0	−12	−91	−34
20	20	4	20	−1	−20	0	−18	−100	−32
30	29	4	29	−1	−29	0	−23	−108	−29
40	42	5	42	−1	−38	0	−28	−120	−24
50	50	6	50	−1	−51	0	−31	−129	−19

**Table 5 sensors-22-09618-t005:** Measurement of flexion of wrist (All angles are in degrees).

Goniometer	Method 1			Method 2			Method 3		
Joint Angle	X	Y	Z	X	Y	Z	X	Y	Z
0	0	1	1	−1	0	0	−7	−83	−35
10	3	11	11	10	3	0	3	−87	−35
20	2	22	22	21	2	0	11	−93	−36
30	1	29	29	29	1	0	19	−100	−37
40	1	39	39	39	1	1	26	−106	−39
50	2	50	49	48	3	1	34	−113	−43

**Table 6 sensors-22-09618-t006:** Measurement of extension of wrist (All angles are in degrees).

Goniometer	Method 1			Method 2			Method 3		
Joint Angle	X	Y	Z	X	Y	Z	X	Y	Z
0	16	NAN	17	−1	16	1	2	−69	−35
10	19	10	22	−11	19	1	−4	−61	−35
20	21	18	29	−20	21	1	−10	−54	−35
30	23	31	37	−29	22	1	−18	−47	−36
40	24	39	45	−41	23	1	−25	−40	−38
50	25	49	54	−50	24	2	−32	−32	−41

**Table 7 sensors-22-09618-t007:** Measurements of pronation of wrist (All angles are in degrees).

Goniometer	Method 1			Method 2			Method 3		
Joint Angle	X	Y	Z	X	Y	Z	X	Y	Z
0	18	NAN	18	−1	18	−1	4	−68	−37
10	22	12	19	−1	19	11	4	−67	−24
20	27	23	17	−1	17	21	3	−69	−15
30	34	28	16	−1	16	30	3	−69	−5
40	42	41	15	−1	15	38	2	−70	4
50	52	51	14	−1	14	52	1	−71	14

**Table 8 sensors-22-09618-t008:** Measurements of supination of the wrist (All angles are in degrees).

Goniometer	Method 1			Method 2			Method 3		
Joint Angle	X	Y	Z	X	Y	Z	X	Y	Z
0	17	NAN	17	−2	17	0	3	−68	−35
10	18	9	16	−1	16	−12	2	−70	−46
20	24	22	14	−1	14	−21	1	−71	−56
30	32	31	11	−1	11	−30	−1	−74	−66
40	41	43	9	0	9	−43	−1	−76	−75
50	50	51	8	2	8	−51	−1	−78	−85

**Table 9 sensors-22-09618-t009:** Knee angle measurements from flexion of the leg (All angles are in degrees).

Goniometer	Method 1			Method 2			Method 3		
Joint Angle	X	Y	Z	X	Y	Z	X	Y	Z
0	0	NAN	1	−1	0	0	0	5	0
10	11	10	4	−1	−4	−10	0	1	−10
20	21	20	6	−2	−5	−20	0	−1	−20
30	30	30	7	−2	−6	−30	1	−2	−29
40	40	40	8	−3	−8	−39	1	−4	−39
50	51	51	10	−4	−9	−50	0	−6	−50

**Table 10 sensors-22-09618-t010:** Elbow angle measurements from flexion of the arm (All angles are in degrees).

Goniometer	Method 1			Method 2			Method 3		
Joint Angle	X	Y	Z	X	Y	Z	X	Y	Z
0	0	NAN	1	−1	0	0	−2	1	−80
10	10	NAN	10	−1	10	0	7	3	−80
20	21	NAN	21	−1	21	0	18	5	−79
30	31	NAN	31	−1	30	0	27	7	−78
40	40	NAN	40	−1	42	0	36	10	−77
50	51	NAN	51	−1	50	0	46	13	−74

## Data Availability

Not applicable.

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
