# Peer review of "Posture Monitoring and Correction Exercises for Workers in Hostile Environments Utilizing Non-Invasive Sensors: Algorithm Development and Validation"

_sensors, 2022, doi:10.3390/s22249618_

Round 1

Reviewer 1 Report

This is a nice article, and the presented methods and sensors are well explained. Some things are explained even too comprehensively, for instance figures 9-11 are not needed.

The problem of the article is the analysis of the results. Now results are in "discussion" and they are analysed in Conclusions. Analyze results in discussion and conclusion should tell the main findings of the article. Moreover, analysis should be improved. Tell how contents of the tables should be understood and what we can learn from them. Especially Tables 4-10 should be analysed in more detail.

Font in figures 12-13 too small. Figure 20 "pressine"?

In text, sometimes figure with small letter and sometimes with capital. Should be capital everywhere.

Improve related work, what kind of IMU-based ergonomy studies have been made?

Reviewer 2 Report

The authors put forward a  filtering and sensor fusion method for pose recognition in hostile environments. The topic is interesting and indeed very hot these years. The presentation is OK but some shortcomings should be improved before it can be accepted.

1. Though the fusion method proposed is OK, how it compares to the data-driven method is barely discussed in the paper. Many public datasets and methods could be accessed, and some discussion should be added in the related work section. I refer the authors to the following reference. Xu C, Chai D, He J, et al. InnoHAR: A deep neural network for complex human activity recognition[J]. Ieee Access, 2019, 7: 9893-9902.

2. The experiment is not so adequate. Some comparisons should be discussed, especially considering state-of-the-art.

Round 2

Reviewer 2 Report

My concerns have been addressed.